# Resource Allocation in Oil-Dependent Communities: Oil Rent and Benefit Sharing Arrangements

**Svetlana Tulaeva [1],\* and Soili Nysten-Haarala [2]**

[1]  Faculty of International Relations and Politics, North-West Institute of Management, Russian Presidential Academy of National Economy and Public Administration, 194044 St. Petersburg, Russia

[2]  Faculty of Law, University of Lapland, 96300 Rovaniemi, Finland; soili.nysten-haarala@ulapland.fi

\*  Correspondence: svett07@mail.ru

**Abstract:** This study is dedicated to the interaction between oil and gas companies and local communities that depend deeply on the production of oil. One of the key concerns of all oil-dependent communities is the distribution of oil rent: Who participates in decision making regarding the distribution of oil profits and who can claim the benefits and on what grounds? Benefit sharing arrangements are used to decide such matters in global practice. Using Russian Arctic and subarctic areas as examples, we analyze the main rules and practices of the distribution of benefits from oil production at the local level. This study focuses on the coexistence of oil companies and indigenous people, many of whom practice a traditional way of life. We also pay attention to the institutionalization of the norms and rules of oil-dependent communities at the local level.

**Keywords:** indigenous people; oil-dependent communities; oil companies; Russian Arctic

## 1. Introduction

The introduction of new technologies in the 19th century suddenly boosted the value of oil, making it one of the symbols of economic progress and political power in the world. From a sticky substance used to light kerosene lamps, it became the key resource for industrial progress [1]. Technological progress and the creation of relevant infrastructure also made oil a supreme resource. By the end of the 20th century, the world had completely turned into a society symbolically referred to as the oil Leviathan [2]. This society has certain distinguishing features. First, there is a close dependence between economic prosperity and the production and export of oil. There is even a group of countries whose progress has depended on the price of oil for many decades [3]. Second, oil can be used as a political tool in the oil-dependent society. The third prominent feature of the oil-dependent society is the development of an oil ideology. A number of countries have started to embrace the idea of national supremacy, based on their control over energy resources. This is why researchers speak of resource nationalism and the use of natural resources as a tool to strengthen the idea of national exclusivity. Resource nationalism, in turn, leads to the development of a power ideology. On the one hand, this provides an oil producing nation with a chance to substantiate its political ambitions in the global arena and to legitimize its existing power or influence and, on the other hand, it justifies a lack of economic growth and material prosperity among the deprived population [3].

Just as any other society, oil-dependent societies have their own sets of rules and norms of behavior, their own beliefs and rituals, their own distribution of power and authority, and their own types of conflicts and the methods for their resolution. According to Douglas Rogers, the depth of the oil well is always directly proportional to the depth of institutional transformations [4]. Most existing research papers on the topic analyze the influence of the oil curse on political institutions at the national level [3,4]. In this study, we focus on the influence of the oil curse on the social institutions of the

oil-dependent society. We also study agreements on the distribution of benefits between oil companies and indigenous people. Indigenous peoples leading a traditional life are the most vulnerable group in terms of industrial development in their territories, and they are occasionally forced to fight for their rights in conflicts with oil companies. One of the key questions regarding such interaction relates to the rules of the distribution of oil rent among various groups of stakeholders.

We mainly pay attention to the rules of the distribution of resources among oil and gas companies, state authorities, and local residents in the oil-dependent communities of Russian regions. The development of the oil-dependent society in the indigenous regions of the Russian North dates back to the Soviet Union. Oil drilling started in the northern territories following the Second World War. Geological surveys were conducted in this area in the 1950s and 1960s, and a little later the identified sources of oil were subjected to drilling. Traditional communities were affected by the construction of industrial facilities, the shrinking native territories, the arrival of many nonlocals bringing along new lifestyles, the development of infrastructure, and the spread of technology [5,6]. Following the perestroika era, a crisis and a shift to a market economy, the oil and gas companies that maintained the infrastructure in these areas became the key economic players. The shift from a planned to a market economy required the development of new rules of engagement between oil companies and local communities. The process of enacting the new rules was accompanied by a rise in the number of conflicts [7–10].

We can identify some of the reasons behind these conflicts. First, the increase in oil production led to a decrease in agricultural land area used by the locals. This problem is very acute for indigenous people, who practice a traditional lifestyle. The shrinkage of land threatened the conservation of reindeer herding and other traditional lifestyles [5,8]. Land-related conflicts were persistent in Yamalo-Nenets Autonomous Okrug (YNAO) in the 1990s and in Nenets Autonomous Okrug (NAO) in the beginning of the 2000s. In Yamal, such conflicts grew into open protests, whereas in NAO they were latent by nature. Second, oil production affected the environment. Industrial activity deteriorated pastures and decimated fish populations in the rivers. Despite the related legal environmental norms, oil production brings along spills and other industrial accidents [9–12]. In fact, the deteriorating environment became one of the reasons why indigenous people protested against oil companies in Yamal in the mid-1990s and in the Republic of Komi in 2010–2016. Third, industrial activity desecrated indigenous people's holy places. Since many indigenous communities adhere to pagan beliefs, it was important for them to preserve their holy lakes and sockets, i.e., the places in which they could perform pagan rituals. Oil drilling close to the holy sites of indigenous people indeed became one of the causes of conflict between oil companies and indigenous people in Sakhalin in the mid-2000s. Exxon Oil and Gas Limited started using indigenous people's holy field to store their industrial stock, which aroused dislike among the local population and became the starting point of a wave of protests against oil and gas companies in Sakhalin [13–17]. A similar incident took place in Khanty-Mansi Autonomous Okrug (KMAO), where a conflict broke out in 2015–2017 because of oil drilling performed by "Surgutneftegaz" in the Numto and Ilmor lake area. Despite the fact that it was a holy territory, the oil company was not willing to give up on these lands, because they held vast reserves of oil [9,18,19]. Last, conflicts were caused by the amount of money paid by the oil companies to local people to compensate for damage. This issue was very acute in the 1990s, when there were no formalized rules for assessing compensation [18,19]. In this paper, we analyze the different ways to resolve such conflicts between companies and local communities in oil-dependent Russian regions. The article focuses on the rules of the distribution of oil rent in oil-dependent communities and the ways in which these rules affect social institutions.

The main questions of this research are the following: (1) How are the rules of oil rent distribution institutionalized at the local level in oil-dependent communities and (2) how does the order of the distribution of oil rent affect societal relations?

## 2. Theoretical Framework: Oil, Institutions and Society

The theoretical framework of this study is neo-institutionalism. According to this approach, institutions are understood as formal and informal rules of interaction between actors [20]. Institutions are created by people to reduce transactional costs and the related uncertainties. Institutions mean cognitive, normative and regulatory frameworks that make it possible to integrate collaborators' actions, to determine the hierarchy of positions and to identify the different stakeholders operating in the same space. They form and sanction a relatively uniform code of conduct for inter-personal relations and help in the mutual creation of a shared system of values [20].

The process of institutional formation is complicated and is influenced by many factors. Of these factors, we focus on the availability of significant natural resources such as oil. Many studies have been conducted on the interdependence between the resource curse and institutions [21–24]. Can the physical nature of a key natural resource determine the political and social fabric of society? The majority of researchers agree that there is a close bond between the abundance of resources, the level of state control, and the development of institutions in less-developed, resource-rich societies. Ch. Tilly and M. Olson have noted that the formation of states was due to the institutionalization of certain forms of management and protection in exchange for the provision of some resources to military commanders [21,22]. In given territories, rulers enabled the formation of political and economic institutions that stabilized local people's lives in exchange to acquiring the right to collect taxes. According to A. Etkind and M. Ross, the nature of the resources available to the state and the rulers determines the formation of appropriate political institutions. In other words, in exchange for resources, the state provides the people with various institutions. Differing resources have differing features that are connected not only with their physical characteristics but also with their institutional characteristics [23–25]. This means that while being one of the key resources in modern times, oil also has specific features that affect the development of political, economic and cultural institutions in oil-dependent communities.

Researchers focus on some of the institutionally significant features of oil that play an important role in less developed, undemocratic societies. One of the key features of oil is its capability to bring considerable profits for the state. M. Ross notes that in forming state institutions, the main source of state revenue plays an important role. To rephrase, a person is what he eats and state is what funds it collects [8]. If the state's main source of revenue is taxation, the state is more considerate toward society and dependent on it. However, if the state can collect revenues from other sources, such as the export of oil, it is less likely to be affected by the pressure exerted by the citizens. The economic non-interest of the oil Leviathan in its citizens is accentuated by the fact that the oil industry does not need much workforce. The production of oil requires a limited number of highly qualified workers. According to A. Etkind, this determines a government's attitude towards the people of the country: citizens turn from a resource and a source of prosperity into objects of philanthropy. The rulers do not consider the people as a valuable resource that needs to be developed as a good investment. Therefore, some of the oil and gas producing countries are super-extractive states, where the prosperity of the political and economic elite does not depend on the people, but is determined by other factors such as global oil prices [23]. In such societies, the main reason why the state subsidizes its citizens is to gain their political loyalty [24–26]. The state buys its citizens' loyalty by increasing social transfers of the oil revenue towards the people [25]. However, the spending of hyper-profits generated by oil sales leads to an increase in secretiveness and translucency. In many oil-producing countries, citizens do not have the capacity to control the amount of money that their government earns through oil sales. In fact, it is practically impossible to find authentic data about the countries' oil revenue. Combined with weak democratic structures, this reduces the social accountability of the government [23].

High-profit resources ferment conflicts over the right of ownership. According to research, these conflicts are more frequent in oil-dependent societies than elsewhere [23,27,28]. The distribution of oil rent in these societies is not equal, and the benefits and costs of oil production are always distributed hierarchically.

Conflicts can be latent or they may turn into armed struggles. Not all researchers agree that there is a clear connection between conflicts and oil. For example, M. Ross notes that oil can have two contradicting effects: it may restrain insurgencies, as it increases people's income, or it may cause revolts in a country's oil-rich regions. If oil restrains conflicts, it affects the whole country, but if it instigates them, only the oil-rich regions are affected [23]. G. Schlee is also doubtful about the influence of natural resources on the level of conflict in society. According to him, there is no such thing as a conflict over resources or a resource-instigated conflict. He maintains that any conflict that looks like a conflict over resources is in fact a conflict caused by ethnic, religious and/or linguistic differences. Oil is often merely a catalyst of conflict [28]. M. Ross supports this view by noting that oil-related conflicts are amplified in areas where oil is complemented by ethnic or religious dissatisfaction [23].

All over the world, the oil and gas industry mainly develops in vertically integrated holdings [4]. Such form of industrial organization enables the reduction of transaction costs and increases the stability of the companies. To guarantee the security of its investment, a business must integrate the source of oil and the refinery into a single industrial structure. Meanwhile, a vertically integrated business structure is based on a strictly hierarchical system of management, which is also applied to the business's social activities. An oil company's social and cultural projects are also executed within the framework of this vertically integrated structure of management [4]. These projects are supposed to legitimize the dominating stakeholders, as well as to create an aura of social prosperity and equal distribution of oil benefits. K. Humphrey equates oil companies with medieval overlords, who took control of large underdeveloped territories with minimum infrastructure and local populations, awaiting comprehensive support and large cash flows [29]. In such cases, social and cultural policies are also subjected to vertical integration and hence become dependent on the major players. The social and cultural projects executed by oil companies and the state in oil producing areas are selected from the perspective of the dominating players, and they do not necessarily conform to the needs of the majority of the people [4].

Hence, many researchers say that oil and gas as key natural resources determine the formation of relevant political institutions and stimulate the advancement of autocratic political systems. It has also been noted that an abundance of oil deteriorates effective institutions or leads to their replacement with less-effective ones [25]. Other researchers tend to support the idea that the oil curse is not a crucial factor of poverty. The slow economic growth in some oil-rich states is not due to the negative influence of excessive oil profits, but to the ineffective institutions that the states have inherited [23]. The oil curse will have negative consequences in a society with weak democratic institutions. In this case, the availability of oil revenues increases the possibility that the authorities ignore the participation of citizens in decision-making processes and use the practices of social patronage towards the citizens. The governments of oil-rich countries are not always ready to resolve problems associated with the distribution of oil rent, the lacking interest of the state and oil companies in the population, and the fluctuation of oil revenues. The authorities tend to mask or negate the existing problems through relevant cultural and social programs. Therefore, it is necessary to pay attention to institutions that regulate the distribution of oil revenues within oil-dependent communities. Examples of such institutions are benefit-sharing agreements between oil companies and local residents. In this paper, we address the institutionalization of these agreements in Russia.

## 3. Research Materials, Methods and Research Questions

This study focuses on the sharing of benefits in the regions of northern Russia where oil companies operate: Nenets Autonomous Okrug (NAO), Khanty-Mansi Autonomous Okrug (KMAO), Yamalo-Nenets Autonomous Okrug (YNAO), and Sakhalin Island. NAO and YANAO are located above the Arctic circle, while Khanty—Mansi Autonomus Okrug is situated in western Siberia and borders the Yamalo-Nenets Autonomous district. Sakhalin Island is located in northeastern Russia (See Figure 1). Oil and gas corporations are operating in all of these regions, where there is very little other industry that would not depend on the oil industry. On the one hand, oil companies build

and maintain most of the local social infrastructure, and on the other hand, they determine the social and political trends in these territories, thereby converting them into oil-dependent communities. Since tax revenues flow almost totally to the federation, the regional authorities are dependent on good relations with the companies. Based on soviet legacy, companies are expected to take care of the local infrastructure as negotiated with the regional authorities. Although this bond between the companies and the regional authorities resembles taxation, it leaves the companies a lot of power to decide the way in which they support the regions [5].

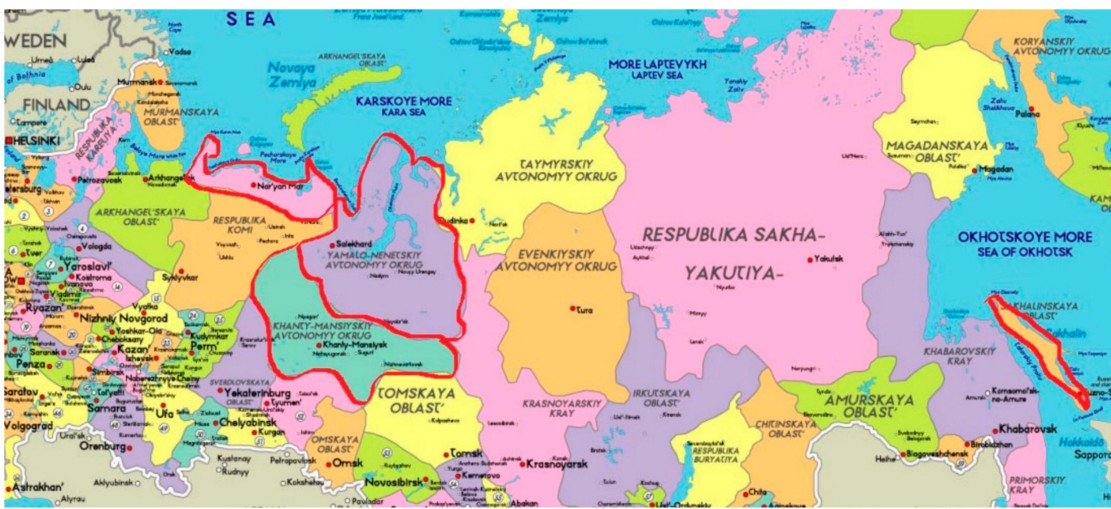

**Figure 1.** Administrative map of Russia. Source: Maps of Russian Federation. URL: http://www.maps-of-europe.net/maps/maps-of-russia/administrative-map-of-russia.jpg.

A distinctive feature of these regions is that they contain large groups of indigenous people, some of which follow traditional lifestyles. These indigenous groups are the Nenets (NAO, KHMAO, YANAO), the Khanty and the Mansi (KHMAO), and the Nivkhi and the Ulta (Sakhalin). Their livelihoods represent subsistence economy: fishing, hunting and reindeer herding. Although there are some markets for their products, their livelihoods cannot compete with the oil industry in economic productivity. People representing this lifestyle do not share the same ideal of economic prosperity with oil companies, regional authorities or other local populations. According to tradition, they are satisfied with living off the land, which they aim to pass to the next generations. The local residents view the arrival of oil companies in an ambivalent manner. On the one hand, oil extraction is regarded as a matter of national importance and an opportunity to receive significant economic support from the companies, which in fact finance the construction of schools, gymnasiums and residential buildings. The material aid that the local residents receive from the companies provides them greater comfort. On the other hand, the arrival of the oil companies is perceived as a threat to the traditional way of life in the tundra, which indigenous people consider their home. For this reason, indigenous people are concerned about the forced limitations on reindeer herding and environmental pollution caused by industrial development. These people have always been closely tied to land; that land is not only important for reindeer herding and fishing, but also the basis for their spiritual culture. The reduction of land area inhabited by indigenous people alters their social norms and deteriorates their ethnic identity. According to our research, all these processes stimulated social transformations in the studied local communities.

This study is based on grounded theory and qualitative methods [30,31]. The use of grounded theory included the following stages: collection of empirical data, their careful coding, selection and interpretation of analytical concepts, comparison of the analytical categories with new data, further elaboration of the analytical concepts, formation of theoretical conclusions, and validation of the results. At the same time, data collection and analysis were carried out in parallel. The main methods

of data collection were semi-structured interviews, document analysis and participative observation. We interviewed representatives of the regional and local authorities, representatives of oil and gas companies, local residents of oil production areas, and experts from scientific and research institutes and social organizations. The interviews were conducted with representatives of the following companies: Lukoil, Surgutneftegas, Sakhalin Energy and Exxon Neftegas Limited. In addition, interviews were conducted with representatives of the state authorities and local residents in Salekhard, Seikha (YNAO), Yuzhno-Sakhalinsk, Nekrasovka, Okha, Nogliki (Sakhalin), Khanty-Mansiysk, Numto (KHMAO), Naryan-Mar, Nelmin-Nos Horei Ver, and Krasnoe (NAO). A special guide was appointed for each group of informants. The crucial issues were focused on the main forms of interaction between companies, local residents and authorities, as well as on the economic and social consequences of the existing forms of interaction for local communities. The respondents were asked the following questions: What forms of corporate and state assistance to local communities are the most widespread? What forms of assistance are the most effective from your point of view and why? What are the negative impacts of industrial activities on local communities? What are the causes of conflicts between companies and local residents? The informants were selected with the help of the snowball method. Altogether, 95 interviews were conducted, and they were all subsequently transcribed and analyzed.

The main types of data analysis were coding, highlighting the main analytical concepts and categories, and their further development and interpretation. The interviews were analyzed through thematic, axial and selective coding. In analyzing the data, we identified some of the most significant codes, which were then combined into larger analytical categories. The key category of analysis was the concept of oil-dependent community. We divided this concept into the following subcategories: norms and rules of oil-dependent communities, myths and rituals of oil-dependent communities, cultural fakes, conflicts and risks in oil-dependent communities, and tools for conflict resolution. For a more complete disclosure of categories, we used memos. During our research, we revisited the selected codes and categories and made new interpretations based on newly collected materials. In further data collection and analysis, we compared the new data and the existing analytical categories, refined them, and made a more complete description and interpretation of the categories. Disclosure of the key category through the constant comparison and the addition of related categories was used to specify the studied social processes conceptually. This allowed us to substantiate our conclusions about the influence of oil resources on societal relations.

Furthermore, we used participative observation as a research method in the following indigenous settlements located in oil-drilling areas: Krasnoe, Nelmin Nos, Horey Ver (NAO), Nekrasovka (Sakhalin), and Seiha (YNAO). The observation was conducted during a period lasting from two weeks to a month in each community. During the observation, we lived in the settlements and took part in the daily affairs of the communities together with the residents: we attended public events in cultural centers, museums and village councils; we went fishing and gathering; and we participated in informal meetings and holidays. While observing, the researchers kept diaries that were later analyzed. This llowed us to study the life of the communities from the inside and helped us to reveal practices of interaction between indigenous communities, oil companies and state authorities. Additionally, it allowed us to analyze the implementation of various rules concerning benefit-sharing arrangements.

We also analyzed the federal and regional laws that regulate the relationship between oil companies and local residents, and studied local press and corporate reports about sustainable development. The following federal and international documents were analyzed: the Land Code; the laws On Subsoil Resources, On the guarantees of the rights of indigenous small-numbered peoples of the Russian Federation and On the territories of traditional use of natural resources of the small-numbered indigenous peoples of the Russian Federation; the Convention on Biological Diversity; the ILO Convention on the Rights of Indigenous Peoples, etc. Furthermore, we analyzed corporate reports and regional media in order to identify the main trends of corporate social programs in the regions.

The data were collected between 2011 and 2017 as follows: NAO (June 2011), Sakhalin (August 2015), YANO (July 2017), KHMAO (January 2016, February 2017) and NAO (2016) (We are very

grateful to Maria Tysiachniuk and Galina Grening for their contribution to the materials). The duration of the study over several years allowed us to analyze the changes that have taken place in the local communities.

We used triangulation to ensure the validity of the interviews, observations, publications and documents. The combination of materials helped us to analyze the practices of benefit-sharing arrangements from the perspective of various groups of actors.

We anonymized the data to protect the privacy of our informants. Each interviewee was informed about the nature of the research and publications, and their oral consent was received prior to participation. The article does not provide facts about the actors, but it does describe the main practices of interaction between indigenous communities and oil companies in Russian regions.

In this research, we focus on the influence of oil on institutions and identify the basic aspects of the changes caused by this influence. We also analyze the transformation of social institutions under the impact of oil revenues. Then we discuss the main stages of oil expansion, the specifics of the distribution of authority and resources in oil rich regions, and the conflicts associated with these processes. Further, we elaborate the main forms of agreement between oil companies, indigenous people and the authorities in the studied Russian regions. We also discuss the norms and understandings that are rooted in oil-dependent communities and are influential in terms of the execution of agreements. Finally, we present conclusions about the basic features of agreements related to the distribution of benefits in the regions.

## 4. The Institute of Benefit-Sharing Agreements in Russian Oil-Dependent Communities

The prosperity of oil-dependent societies is based on oil rent, but the rules of the distribution of oil benefits rouse many differences of opinion and discussions, particularly because the redistribution of oil revenues in oil-dependent societies often occurs on an informal basis. Concurrently, various instruments of governance, aimed at a just and equitable distribution of natural resource benefits among various groups of stakeholders, are actively being promoted [20,32]. That said, the concept of just and equitable distribution of benefits has gained wide acceptance around the world. It originated in international conventions aimed at protecting biological resources and is associated with the idea of social and environmental justice [32,33]. This concept has also been widely accepted in the field of oil extraction [34,35]. The just distribution of resources is implemented in the form of benefit-sharing agreements between oil companies and local residents. On the one hand, these agreements take into account interaction with indigenous peoples, since oil extraction influences their environment and traditional lifestyles. Intensive oil production leads to environmental contamination, reduced wildlife populations, altered migratory routes, and lost fish populations in rivers. This has a negative effect on hunting and fishing, which are traditional livelihoods of indigenous people. On the other hand, the expansion of oil extraction means occupying land areas that are used by indigenous people [36]. Based on all this, indigenous people are entitled to a share in the revenues of industrial production [32,33].

Normally, benefit-sharing agreements are considered in the literature as a positive innovation that promotes sustainable development in natural resource governance. These agreements enable local communities to share the benefits of natural resource extraction. In many cases, the compensation is calculated according to damage and lost profits. Other arrangements include partnership or sponsorship agreements. Such agreements are regarded as a means for indigenous people to protect their rights, to support and strengthen their own institutions and to preserve their traditional lifestyle [34,35]. However, there is no consensus on their effectiveness. From the point of view of corporate managers, the agreements help to establish positive and trustworthy relations between companies and local residents. From the point of view of local residents, companies primarily use these agreements to achieve pragmatic goals and to reduce social and economic risks—in other words, they have nothing to do with trust. A number of researchers are skeptical about these agreements, saying that they do not

allow locals to participate in decision making. If resources are not redistributed justly, the agreements do not prevent inequality [37,38].

Some of the main aspects of reaching such agreements are highlighted in the scientific literature [32–34]. First of all, there are benefits that communities can gain from companies. These benefits can be passed to a community in various forms: as taxes (formal benefits), as compensation (formal benefits), as private-public partnerships initiated by authorities, companies and indigenous people (semi-formal arrangements), as charity programs (charitable giving), and as side benefits of industrial activity, including the creation of jobs and infrastructure development (trickle-down benefits) [39].

The second important aspect of entering such agreements is transparency and the participation of local residents in the distribution of benefits. This involves the introduction of transparent and formalized rules that govern the redistribution of benefits [36,38,39].

Finally, these agreements serve to address companies' degree of influence over local communities. This includes the regular evaluation of environmental contamination, changes in the lifestyle, the degree of loss of traditional culture, and the deterioration of the social fabric caused by industrial activity. It has been posited that the level of compensation to a community from a company is proportional to the effects of the industrial activity on the community [30,38,39]. In this study, we address agreements between oil and gas companies and indigenous peoples in Russia.

Russian practice regarding these agreements differs from international practice. In Russia, the agreements are concluded not only between indigenous peoples and corporations, but also between companies and authorities. It means that the funds received from companies are used both for indigenous people and for the rest of the population living near industrial enterprises. The idea of such agreements is rooted in the Soviet era, when state-owned enterprises were responsible for the social sphere of the settlements in which they operated [5,39].

Nowadays oil companies, locals and authorities use various types of agreements to resolve conflicts. These applied agreements enable compensating the locals for damage and supporting the local communities. They function as benefit-sharing agreements in the Russian context. The main types of benefit-sharing agreements used in the Russian regions are as follows [5,18,19,39] (See Table 1):

- 　Semi-formal agreements on compensating for damage;
- 　Formal agreements on compensating for damage;
- 　Partnership agreements for socio-economic collaboration;
- 　Sponsorship agreements.

**Table 1.** Types of benefit-sharing arrangements in the Russian regions.

| Form of Agreement | Basis of the Agreement | Transparency | Decision Making | Focus of the Agreement |
|---|---|---|---|---|
| Semi-formal agreements between companies and indigenous people | Personal relations | Non-transparent | Companies, state | Material support |
| Formal agreements between companies and indigenous people | State legislation | Transparent | Companies, state | Financial compensation |
| Partnership agreements between companies, state, and indigenous people | Corporate and international standards | Transparent | Companies, state, communities | Development of local initiatives and social infrastructure |
| Sponsorship | Corporate standards and personal relations | Semi-transparent | Companies | Small charitable donations |

*Semi-formal agreements between oil companies and indigenous people on compensating for damage caused by the oil industry.* These agreements were typically concluded by and between companies and indigenous communities without the participation of the state. According to these agreements, the companies

would pay reindeer herders compensation for land annexed for oil production. The money was used for vehicles, petrol, processing equipment, food, and transportation. Most of these agreements had been made on a short-term basis. The contents of the agreements were confidential, and only community leaders usually knew the amount of the money paid. Agreements were also often made with municipalities. In Russia, they are legal persons and their populations often consist of both indigenous and non-indigenous people. Sometimes oil companies also made agreements with the leaders of indigenous farms.

The amounts of compensation were determined in informal discussions between the oil companies and local leaders, and they mostly depended on the negotiation skills of the latter. If the locals did not have the relevant knowledge and skills, the amounts of compensation were insignificant. *"Oil companies signed support agreements, for example to transport you to and from, and you would sign the agreement that I will pass through your lands. Like in the case of Indians, precious wealth was plundered"* (Indigenous person, KMAO, 2017). Such agreements were very common from the 1990s up to the mid-2000s, when active oil production started in the northern territories. Both parties noticed the instability of such agreements: *"We are building an oil pipeline, for land we have already paid, but he says: Give me three more tons of diesel. I say: Listen, we have already paid—and he says: you give me more right now, I need it. I say: you're insolent. He: Well, then I will not let you build"* (Representative of an oil company, Komi Republic, 2015). Currently such informal agreements have been replaced by more formalized calculations of compensation.

*Formalized agreements between oil companies and indigenous people on compensating for damage caused by oil drilling*. These are agreements on compensating for damage in favor of indigenous people in connection with the acquisition of land and lost profits. The agreements are signed between oil companies and indigenous households, indigenous enterprises or indigenous NGOs. They are based on Russian Laws passed during 1990–2000 and aimed at regulating the relationship between oil and gas companies and local people. The manner of signing them varies from region to region because of disparities between regional laws. For example in KMAO, the law Territories of Traditional Nature Use (TTNU) was passed. According to this law, indigenous people who have registered rights to territories of traditional nature management, can receive substantial compensation from oil companies that engage in industrial activities on their land. If an indigenous people do not have officially registered rights to such a territory, then the companies will sign the relevant agreements with the local administration that will collect the compensation. In such cases, the local administration decides how and when to spend this money [39]. In Yakutia, there is a law on ethnological impact assessment that also allows indigenous people practicing a traditional lifestyle to receive compensation from companies. In some regions, indigenous people have not officially registered as owners or lease holders of land and therefore cannot count on receiving compensation from companies. In such cases, only reindeer farms with legally registered rights to agricultural land are entitled to compensation. For example in YNAO, only state reindeer enterprises have officially registered land areas, while ordinary reindeer herders do not, and therefore cannot, receive compensation.

The amounts of compensation are determined in accordance with a formal method, approved by the Ministry of Economic Development in 2009, of calculating the damage when agricultural land is acquired for industrial use. This method is applied in YNAO and NAO. In comparison with the previous practice of concluding semi-formal agreements, the calculation of compensation in accordance with the formal rules proved to be more beneficial to reindeer herders: *"The previous method was less expensive for the companies ... which means that the digits were fewer"* (Representative of an oil company, Komi Republic, 2015). The use of formal rules of calculating compensation also enabled companies to hedge themselves against additional demands from reindeer herders, and to reduce the risk of conflicts with locals: *"Transitional period—troubled times of reformation—you could ask for something. Since the interaction is limited within the bounds of the agreement, you cannot demand anything. The companies consider such asking as blackmail"* (Representative of local administration, NAO, 2016). However, the new rules underwent amendments. In 2017, the method of calculating losses caused by the annexation of agricultural land was amended, which reduced the amount of compensation.

According to the interviews, the new way of calculating these payments became more profitable for oil companies, while indigenous communities lost some money (Director of reindeer herding enterprises, YNAO, 2017).

Apart from agreements on compensating for the annexation of land and lost profits, there are other compensation agreements, albeit less significant ones. These include agreements of servitude, which allow the use of agricultural land for industrial purposes without changing the category of the land. An example of such an agreement is the use of agricultural land for constructing a winter road to be used by an oil company.

*Partnership agreements between companies, authorities and indigenous people.* Partnership agreements can be bilateral or trilateral. Normally, these agreements are concluded at the regional or district level. The agreements are voluntary, since there are no legal requirements to sign them. However, in practice, refusal to sign one will result in troubled relations between a company and the authorities, as well as in bureaucratic hurdles: "*You cannot enter a territory without charitable or sponsorship spending*" or "For any incoming company, it means a school or kindergarten" (Oil & Gas company's representative, Republic of Komi, 2015). Within the framework of these agreements, companies allocate funds for the construction of social infrastructure in villages. Such agreements can be aimed at supporting the whole population or the indigenous people in the oil drilling area. They are used to build schools, hospitals, sports facilities, cultural centers and residential buildings and to support local entrepreneurs. There are no formal criteria for the amount of money the company has to pay, but normally to the sums correlate with the amount of oil produced in the region. In most cases, the regional authorities decide how and when to spend these funds.

An exception to the above is the island of Sakhalin, where Sakhalin Energy and Exxon Neftegaz Limited attempted to introduce international standards that govern community relations. Following the demands of international organizations, Sakhalin Energy decided to assist the development of indigenous communities by drawing up a plan for supporting the indigenous peoples of North Sakhalin. This plan comprised a trilateral agreement between Sakhalin Energy, the regional authorities and the Regional Council of Authorized Indigenous People and it was aimed at strengthening the company's image as a socially responsible operator and at establishing a partnership with the indigenous communities [40,41].

The key idea behind the plan was to develop a partnership with the community. It was based on international standards sanctioned by the IMF, the World Bank, and the ILO [36,37]. It was also influenced by the corporate standards of Shell and BP. The company attempted to create a transparent and collegial decision-making procedure. The plan was to be managed by the company and the representatives of the local authorities, and the main decisions regarding the distribution of funds were to be made by the representatives of the indigenous peoples [40,41]. Exxon Neftegaz Limited, based on the experience of Sakhalin Energy, developed a similar model of interaction with the indigenous communities. The company made a trilateral agreement with the regional authorities and the indigenous peoples. However, the main decisions regarding the distribution of funds to social projects were made by the company managers with the contribution of representatives of the indigenous peoples and authorities.

*Sponsorship.* Through this type of an agreement, a company provides small-scale financial support to local communities. The support consists of financing educational and cultural events, gifts for veterans, transportation, training etc. Within the framework of these agreements, companies also provide small amounts of money (on average 30–50 thousand rubles) to local libraries, schools and cultural centers.

The abovementioned agreements make it possible to reduce conflicts between companies and local residents. They also enable the redistribution of a small portion of profits to communities. In Russian regions, conflict resolution and the signing of benefit-sharing agreements takes place mainly within a framework that is typically found in a neopatrimonial society. The local peculiarities of this activity will be discussed hereunder.

## 5. Findings

*5.1. Oil and Local Neopatrimonialism*

Authoritarian relations in oil-dependent societies are particularly patrimonial by nature [8,10,11]. They mainly exploit political and economic authority for personal gain through an overwhelming body of informal rules and exchanges. A strictly vertical system of relationships and the presence of a dominating actor are a prerequisite for neopatrimonialism. The accelerated growth of neopatrimonialism in oil-dependent communities is related to the capabilities that authorities gain through oil money and the ensuing limitations. Authorities are given a possibility to generate hyper profits without substantially reforming the economy and society. Meanwhile, they need not have any economic interest in the local residents because of the confidentiality of hyper oil profits and a nontransparent system of profit distribution [8]. Interaction between oil companies, authorities and indigenous people typically exhibits the following features.

First, the interaction is based on a hierarchical model that entails the domination of a certain group of actors in the decision-making process, while the rest of the stakeholders try to adapt to the dominant actor's disposition. At the regional level, oil companies and authorities are the dominant actors. The rest of the actors attempt to establish good relations with them and to seek their support. This enables them to attain financial support for arranging events in villages, purchasing necessary equipment for schools and hospitals, paying for air travel from settlements to the regional centers, organizing events in cultural centers and libraries, and constructing residential buildings.

Second, this interaction is personalized and its success depends on personal contacts. Formal rules are not as much of a key factor in such interaction as are established contacts: "*Their (company's) director has changed, so we need to go and establish contact. The first contact is the most important one. If he gets a good feeling about us, he will give us something. Earlier, we used to take the longer path. But when you go straight to the director, everything is decided very fast. It all depends on whether you can establish a good relation with the director. Go there to decide on some matters: overflows and other such things*" (Director of a reindeer farm, NAO, 2017).

Third, the relationships are based on semi-formal exchange. This takes place among various groups and helps distribute the oil rent among them. Despite the fact that oil companies get their main permits and licenses from the federal authorities, some matters of influence, related to bureaucratic processes, are in the hands of regional authorities.

Overall, in order to succeed, companies must obtain regional support. To gain it, they are required to make significant social investments, which leads to an overall improvement of the economic and social climate of the region. Semi-formal exchange between oil companies and authorities enables social patronage practices that benefit the majority of the population. It also helps to maintain economic prosperity and political stability in the regions: "*Let's do this and that. The governor has signed a new agreement on the first of May. Under this agreement a lot of money was allocated for road repairs in the city, since soon we will have elections, hence people might show annoyance*" (Representative of an oil company, KMAO, 2014).

The fourth feature is the variability of formal rules concerning the redistribution of funds. In a number of cases, the presence of formal rules is not the only basis for making decisions. Until the end of the 2000s, the amount and form of compensation to indigenous people was determined during the course of informal negotiations between the local leaders and company managers. The outcomes of such negotiations could be unknown to both the authorities and the members of the community. However, the existence of formal rules does not necessarily change the situation. If the rules contradict the interests of the dominating stakeholders, they are either revised or adjusted when applied. For example, issuing a method of calculating the amount of compensation when annexing agricultural land for industrial purposes did formalize the interaction between companies and reindeer herders. However, a few years later—in 2017—the method was revised, which led to a reduction in the amount of compensation. In many cases, despite the legally affirmed rights of indigenous people to their traditional territory,

they usually cannot stop oil companies from using the land for oil production: *"They are obligated to get our consent under 145 regional laws. We cannot say yes or no. As per their understanding, we must always say yes. If we say no, they start pressuring us"* (Indigenous person, KMAO, 2014). For example, in a conflict between locals and oil companies in KMAO caused by oil production in the area of Lake Numto, the company initially tried to use formal tools of conflict resolution. They conducted open hearings and considered various options of compensating the indigenous people. However, when the locals refused to accept any form of oil production in the area, the company lobbied its interests through the regional authorities, who shared the values of economic prosperity with them [34,38].

Overall, the neopatrimonial system of relations affects the distribution of funds among companies, authorities and indigenous people and influences the social and cultural practices of local communities. In the following, we will take a detailed look at this influence by discussing the cultural consequences of agreements between oil companies and indigenous people.

*5.2. Traditional Culture and Petroleum Patronage*

Social patronage, inherent in oil-dependent communities, is best depicted in the field of culture. Oil and gas companies often participate in the financing and support of various projects related to preservation of the traditional culture of indigenous people. This corporate support can be considered as a form of benefit sharing. It helps companies to convince their western partners of how well they comply with international standards regarding indigenous people. Apart from that, even minor spending on cultural events enables them to gain the approval of indigenous residents. This support includes a number of typical features. The most common cultural programs are related to the financing of traditional festivities [42–44], where indigenous people hold national sports competitions, arrange folklore performances, have national dress competitions, and perform ancient pagan rituals. During these festivities, also meetings between the authorities, companies and locals are arranged. The most popular of these events is the Reindeer Herder's Day. Companies assist in the arrangement of this occasion by covering transportation costs and by purchasing necessary goods, gifts and prizes.

Another important target of financial assistance from oil companies is the preservation of traditional crafts. This assistance is offered to master's and regular courses at schools or cultural centers in wood carving, weaving, embroidery etc. On these courses, students make for instance national dresses, traditional toys and crockery, and items used in performing pagan rituals. According to educational programs, the main objective of such courses is the acquisition of traditional knowledge. Meanwhile, a number of experts note that the function of the courses is to convert knowledge into folklore, rather than propagating its practical applications [45]. The students are familiarized with typical elements of national culture that were created in the past and adhere to a single uniform standard. Often, however, people have already lost the skills needed to prepare such items. In such cases, some items must be ordered from masters in other regions so that they can be placed on display in cultural centers and regional museums.

Companies also give financial assistance to national language preservation projects. The money received is spent for example on printing books containing fairytales and songs in national languages.

Interestingly, representatives of oil companies are convinced that such projects help preserve indigenous peoples' traditions that existed before the Soviet era. In fact, most of these projects are aimed at the reproduction of cultural practices created by the Soviet authorities. For example, indigenous people's fairytales and songs could not even be published without the alphabet that Soviet authorities created for them. Another example of Soviet influence is the widely known Reindeer Herder's Day, which is often presented as an indisputable proof of the preservation of ancestral tradition. However, it is actually a Soviet invention, and was celebrated for the first time in 1932. An article entitled *Down with the Old Holidays*! in a 1932 issue of the Soviet newspaper *Nyaryana vynder* reads: *"Earlier, the Nenets celebrated old holidays . . . But a new life has begun. Today, Party and Komsomol cells were formed here for the first time; they held a meeting and decided not to celebrate old church holidays in the tundra. Instead of Elijah's Day they will now create their own holiday on 1 August, which they have named "Reindeer Day." This is a*

*new rational holiday. We should support it and make it a holiday for all the Nenets people of the okrug"* [46]. The Soviet holiday was dedicated to the profession of reindeer herding. This suggested that reindeer herding was perceived as professional activity, not as a lifestyle practiced by the nomadic Nenets for centuries. As seen from the newspaper article, the goal of this holiday was to abolish old, partly Christianized pagan traditions and to promote new values.

The preparation of traditional items has been adjusted to make them more conspicuous and to bring the indigenous lifestyle to the attention of oil companies and authorities, in particular. Therefore, traditional festival costumes are sewn in vivid colors to attract the attention of the audience. As some regions have already lost the knowledge and skills needed for preparing traditional items, they are ordered from other regions and from people who may have nothing to do with the culture in question: *"On Reindeer Herder's Day we now wear stylized clothes. For example, the deerskin parka has decreased in size. Now we call them miniskirts. Why? Because it has to look spectacular on stage"* (Local resident, KMAO, 2014). Most of the cultural plans are aimed at preserving traditionalism by creating folklore ensembles, reviving old crafts, publishing old tales and legends, arranging national festivities etc.

A few main issues determining the nature of cultural projects supported by oil companies in the Russian regions can be identified. The first and most important one is related to the goals sought by the companies when they participate in the cultural programs of local communities. Rather than developing and preserving local culture, such corporate programs are focused on minimizing the risks bearing down on the company's activity in the region [35]. From the company point of view, cultural events serve to legitimize oil companies in the eyes of local communities and thereby to reduce the likelihood of conflicts [4,12]. Focusing on national culture helps oil companies to create a historical justification for their operation, to relate modern industrial activity with the locality's history. For locals, these events are an opportunity to draw the attention of the stronger actors—authorities and businesses—to local problems. All the while, companies emphasize their financial support in carrying out such events and thereby show their social responsibility. This instrumental approach to culture is typical for both authorities and corporations. Therefore, many projects in the sphere of culture have little to do with culture, and are instead cultural charades that make local communities shift their attention from more acute issues to cultural ones.

The second important issue involves assuming a paternalistic attitude towards indigenous people [34,39]. This concerns state policies as well as corporate projects. In an attempt to support local culture in Russian regions, oil companies reproduce the same practices as Russian authorities typically do. In most cases, corporations prefer to interact not with local people, but with state authorities, providing them with funds to develop the regions at their own discretion. The authorities, for their part, then distribute the gained funds using traditional command-and-control methods. The paternalistic attitude towards grantees is reinforced by a lack of company interest in local residents. Local people, who lack the necessary qualifications and skills, are of no interest to companies as potential employees. Therefore, companies establish charitable relations with indigenous people. As a result, charity programs do not help in balancing the social scene or in fostering culture. Even when oil and gas companies attempt to apply a project approach towards the implementation of social and cultural undertakings in the regions, it does not change the situation much. Despite the fact that the project approach enables the development of local initiatives, the selection of projects is based on a traditionalistic approach to culture.

Third, a definite limitation in developing and selecting cultural programs is oil companies' stereotypical thinking regarding indigenous culture. One of the main stereotypical ideas concerning indigenous culture is that it is an unchanging lifestyle. Despite the fact that many indigenous peoples still engage in traditional activities related to nomadic reindeer herding, fishing and hunting, they regularly use modern equipment, such as satellite phones, diesel-powered generators, snowmobiles etc. This astonishes nonlocal stakeholders. *"They already can't imagine themselves without snowmobiles. And this is no traditional lifestyle. The traditional lifestyle is the reindeer sleigh. And no diesel stations. [Laughs.] Because the traditional lifestyle is the way your ancestors lived. Of course, they didn't have anything*

*like that"*(Manager of an oil company, KMAO, 2014). External actors arrange their expectations and demands regarding a given culture on the basis of stereotypes [42]. The only projects that attain support correspond to the dominating stakeholders' ideas about traditional culture. Hence, when planning cultural programs, indigenous people consider not only the needs of their own community, but also the expectations of external actors sponsoring the events.

According to R. Douglas, a community's dependence on oil production influences its culture. In other words, the more a community depends on oil-related revenue, the more its culture depends on the provider of oil rent. This prompts it to apply a vertically integrated structure, through which it loses space for creativity [4]. Broadly speaking, the vertical integration of culture into the economy leads to the formation of an aggregate of cultural values that propagates all over the country. This propagation is due to broader social processes and the adoption of a broader set of norms and values inherent in an oil-dependent society. It also gives birth to new myths and rituals in society.

*5.3. Myths and Rituals of the Oil-Dependent Society*

The effectiveness and use of formal rules governing the distribution of benefits between oil companies, authorities and indigenous people are mostly determined by the values and norms prevailing in a given society. The specifics of political and economic systems designed to support oil production influence the spread of certain myths and rituals in society. The terms "myths" and "rituals", however, do not necessarily translate into either fabulousness or ineffectiveness. First of all, these terms indicate one of the basic functions of the conducted events—the legitimization of values in a given society. Second, they highlight faith in certain events and procedures. Institutional theory deals with institutional mimicry, which means that the more successful institutional experiments are, the more actors elsewhere copy or reproduce them. In so doing, the initial idea behind the effective rules is lost and institutional reproduction remains only a ritual. In the following, we discuss some examples of institutional mimicry.

The fundamental beliefs of an oil-dependent society sprout from faith in technology, from the predictability of risks and from the presence of a uniform pyramid of values. An analysis of corporate reports and state documentation, available at the Ministry of Economic Development and Department of Natural Resources websites, indicates a focus on the development of safe and efficient technologies for the production and processing of oil. In most of the interviews with authorities and representatives of oil and gas companies, the phrase "*All risks are calculated*" is stated. But in practice, many of the calculations behind the phrase have been made without considering all possible consequences. Such calculations did not help retain fish populations in northern rivers or prevent oil spills and other industrial accidents. The belief in minimizing risks is closely connected to the conviction that all aspects of human activity can be converted into quantitative indicators and monetized—from the effects of oil drilling on soil and water all the way to the value of human life. Some believe that damage can always be first calculated with a certain method and then paid for. However, the paid amounts are not always comparable to the realized loss and cost. Apart from that, the extent of compensation is determined in federal and regional laws, on which the stronger stakeholders often have more influence than the weaker ones.

The beliefs prevailing in oil-dependent societies also lean on widespread cross-hierarchical values, according to which material prosperity outweighs the ecological aspects of oil production and the values of local culture. The basic assumption is that prosperity has the same meaning for all stakeholders. For example, it is taken for granted that all local residents will appreciate new roads. However, among reindeer herders and other traditional communities, people may find the construction of roads alarming because it will allow nonlocal stakeholders easy access to their land, hunting grounds, reindeer farms and holy places. Another example demonstrating the difference between the hierarchical values of oil companies and indigenous people concerns the annexation of land. Local residents practicing a traditional lifestyle often consider authority over their ancestral territories more important than financial compensation from oil companies: "*Many can be bought out, but many don't, because what will*

*we leave for our future generations? Do we want them to say: "Our grandfathers sold everything"* (Indigenous resident, KMAO, 2014).

Another belief among oil-dependent communities manifests itself as faith in the infinite financial capacity of the oil industry. The majority of people in Russia think that oil companies, if willing, can resolve all material problems in the territories where they do business: *"Those who have oil companies in their area are the lucky ones"* (Chief of a reindeer farm, NAO, 2016). However, people seldom realize that the budgets of oil companies have their limitations and cannot cover all the needs of the people.

This belief is supported by the rituals, of which there are two main groups in oil-dependent societies: rituals of public participation and rituals of demonstrating the credibility of calculating and evaluating risks.

The first group of rituals consists of activities such as public hearings and consultations with people, provision of feedback to industrial concerns and authorities, public declaration of standards of social responsibility, creation of coordination councils, and publication of corporate policies and reports. The second group of rituals consists of conducting examinations and audits. Russian legislation stipulates for a mandatory environmental impact assessment (EIA) procedure. Apart from that, companies that function under international standards and are backed by international financial banks must conduct additional risk evaluations to meet the requirements of their international investors. It is to be noted that instead of being a waste of time, arrangements such as public assemblies, expert evaluations and coordination councils actually enable local residents to take part in decision making and in assessing certain risks. These arrangements often promote the local residents' interests and help to avert possible conflicts. However, their effectiveness should not be overestimated. In a number of cases, public hearings are little more than an imitation of public participation and the evaluations are erroneous. Public meetings sometimes concentrate on formalities instead of discussing project details with the locals. In such cases, the hearings may be held in such a way that the locals either do not know about them or are unable to participate because they live far away and cannot travel to the venue: *"We don't demand millions. We just want to be left with some land. I ask them not to drill here, but they insist. They have occupied the territory and we are as if in a cage. They don't back off. You cannot say NO"* (Indigenous resident, KMAO, 2017). On the one hand, the meetings help to take the local views into consideration and to avert the negative impacts of industrial activity. On the other hand, extreme confidence in calculating all conceivable risks and the ritualization of the public procedures reduces local residents' vigilance when faced with the industrial use of natural resources.

## 6. Conclusions

The dependence of society on a natural resource affects the development of economic, political and social institutions. The dependence also correlates closely with the rules and practices prevailing in this society [8,9]. Hence, oil as a key resource affects various changes taking place in society. The availability of oil and the associated super-profits make it possible to maintain a neopatrimonial system of relations in society and to reduce the dependence of the authorities on society. At the local level, this is manifested in a hierarchical system of relations based on access to oil resources, the opacity of the decision-making process and the variability of formal rules governed by the dominant players. Supported by oil-based wellbeing, the social system stimulates the ritualization of procedures related to public participation in decision making and strengthens informal exchange. Oil and gas companies act as agents of oil-dependent communities and transmit the relevant values and beliefs to the people. This leads to a transformation of norms in local communities that start to adopt new behavioral patterns and to form ideas based on a belief in the predictability of risks and the unlimited financial capacity of oil companies. Currently, traditional livelihoods such as reindeer herding, fishing and hunting no longer provide sustenance for locals. Meanwhile, subsidies from the oil industry and the state help to keep villages alive. Intensive oil production and the spreading market economy lead to a future departure of indigenous people from the notion of measure of necessity, which has been a key value of traditional societies. Owing to this change in values, nature is becoming primarily a source of material

prosperity. This stimulates the formation of a universal pyramid of values dominated by economic priorities. People's faith in the financial capacity and assistance of oil companies makes it difficult for local communities to develop their own projects.

It is important to note that natural resources per se do not have any positive or negative connotations. The outcome of their use is determined not so much by the nature of the resource as by the institutional structure of a given society [8]. Insufficient development of institutions of civil participation and an abundance of informal interaction leads to asymmetric access to valuable resources and unequal distribution of the resulting benefits. When indigenous people are not heard, it will inflict considerable damage upon their values and culture. This may cause discomfort among the population and lead to conflicts. One of the key questions regarding conflicts pertains to the amounts as well as the terms and conditions of the distribution of oil revenues. Conflicts have often been resolved merely by allowing oil companies to determine the amount of compensation instead of amending the applicable rules and incorporating the local residents in the decision-making process. Exceptions to the common practice are mostly the result of pressure exerted by external actors rather than the effectiveness of local communities. For example in Sakhalin, indigenous people gained the right to participate in the distribution of grants from the oil and gas companies Sakhalin Energy and Exxon and to engage in the environmental monitoring of their activities. However, the rules were changed in this case because of pressure exerted by foreign investors Therefore, it would not be justified to conclude that conflicts related to oil rent contribute to the construction of a civil society.

Nevertheless, these conflicts forced the state authorities to draft relevant laws that protect the rights of indigenous people. On the other hand, they also forced corporations to pay more attention to interaction with the communities. This resulted in the institutionalization of an arrangement referred to as the Benefits Sharing Agreement. The neopatrimonial framework of power relations in Russian society determines the general trends of institutionalization regarding such agreements. In Russian practice, these agreements take different forms, ranging from semi-formal agreements between companies and local communities to formal compensation and partnership programs. The agreements have both positive and negative effects. The ways in which the agreements are implemented affects the further development of local communities (see Figure 2).

On the one hand, the agreements enable local communities to receive material compensation from companies. This compensation is used by residents to purchase equipment and technology. It also allows them to increase the profitability of their farms and to engage in traditional economic activity in the market space. The development of social infrastructure in local settlements, carried out within the framework of partnership agreements between the authorities, companies and local communities, leads to improved quality of life. Residential houses, schools, hospitals, sports complexes and houses of culture are being built in villages. On the other hand, great expectations concerning financial assistance and compensation to be received from oil companies increases the dependence of local communities on oil money and forces the residents to follow the preferences of the dominant players in the development of social and cultural programs. Oil patronage in the field of culture leads to the development of souvenir cultural practices conforming to the needs and expectations of the dominant players.

In the Russian context, benefit-sharing agreements have specific features. First, they are controlled by the state. In the Russian regions, state authorities play an important role in the signing of agreements. For example, it is the authorities who negotiate with companies about the amount and forms of support they are to give to communities on the regional and local levels. Even when an agreement is concluded between a company and a nongovernmental organization that protects the rights of indigenous people, the negotiations are greatly influenced by the state. As a result, the values of the oil Leviathan dominate the negotiations. Economic prosperity and social benefits overrule environmental concerns and the values of indigenous cultures.

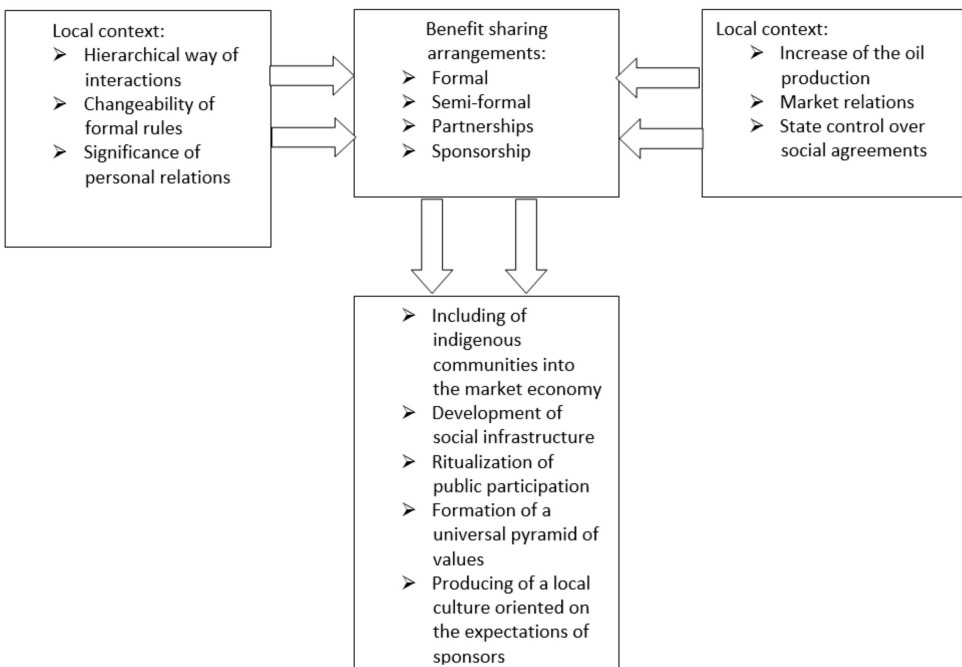

**Figure 2.** Benefit-sharing arrangements in oil dependent communities.

Second, despite the fact that in the last decade the relations between oil companies and indigenous people have been formalized, informal rules that tend to promote the interests of the stronger stakeholders still play a major role. In such situations, the formal rules that should reduce the negative effects of oil production on society and the environment can turn into rituals and lose their significance. Despite the existence of formal rules, indigenous people are often stripped off any possibility to participate in the decision-making processes regarding their residential area. The majority of decision makers and other stakeholder groups share the values of the oil Leviathan and think that the prosperity of their region stems from the oil industry and that any damage to the environment or loss of traditional culture can be repaired with money.

Third, when formal rules are implemented in accordance with their essence, it can lead to unforeseen consequences. In other words, such procedures as risk evaluation and audits must be focused on the impacts of oil production on local communities and the environment. Otherwise they can produce a "Soporific effect", and a situation may arise where a stern belief in the effectiveness of these procedures eventually decreases public concern over adverse industrial impacts.

Further research may provide a more detailed analysis of the social and cultural consequences of economic compensation in oil-dependent communities. That said, the role of compensation agreements in conflict resolution is an interesting topic of investigation. On the one hand, compensation can neutralize contradictions between companies and communities. On the other, competition for corporate support can increase conflicts within communities. In addition, material compensation does not always help to resolve contradictions between groups of actors with contradicting values.

**Author Contributions:** Both authors participated in the development of the theoretical framework of the study, the collection of empirical materials, data analysis and work on the text.

**Funding:** This research was supported by the Academy of Finland's Arctic Academy Programme ("Oil Production Networks in the Russian Arctic," No. 286791) and Fulbright Arctic Initiative 2018–2019.

**Conflicts of Interest:** The authors declare no conflict of interest.

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
