# Peer review of "Resource Allocation in Oil-Dependent Communities: Oil Rent and Benefit Sharing Arrangements"

_resources, doi:10.3390/resources8020086_

Round 1

Reviewer 1 Report

This article is well written and interesting. That being said, I just have a few comments: A map of the Russian regions would be helpful. 

The methods are explained as qualitative but this statement should be furthered categories such as case study methodology, semi-structured interviews, participant observation, content analysis and so on.  

The methods section should further explain how the data was analyzed. Are you using a grounded theory approach or some other? 

The article could be better organized by adding a literature review heading, before the methods section and then after the methods section a findings section, where quotes from the interviews and themes are discussed. 

What the article contributions to theory in the conclusion in particular could be made clearer. The conclusion ending in the bullet points is abrupt. I would either incorporate this as a text or include a final paragraph. 

Author Response

Dear reviewers!

Thank you very much for your fruitful recommendations.  We corrected our article in accordance with your advices. We hope it allowed us to improve our paper.

1.     We included a map of the Russian regions to provide information about Russian regions in a more visible form.

2.     We expanded the methodological part of the article. We explained more carefully the methods of collection and analysis of the data. We described the main groups of our informants and mentioned localities of the interviews. We put some examples of the main questions from our guides. We included brief descriptions of the regions.

3.     We have changed the structure of the article and removed the methodological part of the research after the theoretical framework.

4.     We added analysis to the conclusion and outlined the possible dimensions for further research.

5.     We included two schemes demonstrating the forms of the benefit sharing arrangements in Russia and the influence of the local context on these agreements.

6.     We emphasized the positive and negative aspects of benefit sharing agreements in the conclusion.

7.     We emphasized the social and cultural consequences of economic compensations paid by companies. Two chapters of the article (Myths and rituals of oil dependent societies; Traditional culture and petroleum patronage) are devoted to the influence of such economic agreements on social and cultural norms of the communities.

We are still going to send the draft paper to English proof reading.

Reviewer 2 Report

The submitted manuscript is dedicated to the topic of sharing effects of oil drilling between oil companies and indigenous people, a process of institutionalization of norms and rules in oil-dependent indigenous communities in Russia as well as an influence of this process on their life style. This topic is highly important from the economic, social, cultural and environmental point of view. The question behind the presented research is how this sharing of benefits and losses of oil drilling is compatible with the philosophy of the traditional life style oil-dependent indigenous communities and their interpretation of the progress and prosperity.  However, this study is predominantly descriptive, it deserves more analytical approach and some synthesis.

Thus, the manuscript needs major revision.

The problem of benefits sharing of oil drilling is described really in a complex and objective way; however, the discussion of concept of prosperity and its meaning for different researched groups (oil companies’ representatives, state officials, local non-indigenous persons and indigenous peoples) is missing there. The main benefits sharing concept depicted in the manuscript is financial compensation,  but many indigenous ethnics use to have problem with commerce of natural resources and their interpretation of prosperity differs substantially from the dominant society (as mentioned once in the manuscript). In this context, the discussion of prosperity meaning is really important for interpretation of results.

The situation of oil drilling in the indigenous territory is not just about the benefits shared by oil companies with indigenous peoples but also about losses (damages) shared by indigenous peoples with oil companies. Thus applying of externalities concept in this discussion would be helpful.

In the methodical chapter, the conceptual and theoretical background of the research should be enhanced (brief introduction of theories and theoretical concepts, on which the research is based, has to be provided). The aim of the study and explicit research questions should be described there. It should be explained more precisely there, which method of qualitative research was applied and the concrete application of this research methods should be described in detail, e.g. the application of the (participative?) observation method. In addition, author should provide the list of leading questions used for interviewing informants. The research localities (region) description (the very basic environmental conditions, basic geography) is missing there.

The discussion and synthesis of research findings should be structured somehow, e.g. it could follow the main negative and positive economic, social, cultural and environmental aspects of presented types (agreements) of the coexistence of oil companies and indigenous people. Design of some comprehensive scheme / model illustrating the key findings regarding the sharing both positive and negative effects of oil drilling between oil companies and indigenous people, a process of institutionalization of norms and rules in oil-dependent indigenous communities in Russia as well as an influence of this process on their life style while distinguishing the different perception of prosperity by the oil companies and indigenous people would be very helpful.

In the conclusion chapter, the appropriateness of the research method applied to the study should be discussed. The proposal of some topics or approaches to the related future research is missing.

The citation style could be improved (by mentioning just surname of the cited author and always indicating the number of the given resource).

Author Response

(The authors gave the same response as above.)

Round 2

Reviewer 2 Report

The quality of the manuscript is improved, however, it still needs a substantial revision.

The chapter “Theoretical framework” should be enhanced, the brief introduction of theories and theoretical concepts, on which the research is based, are still missing there (is this environmental economy, cognitive anthropology or what?).

In the chapter “Research materials and questions”, it should be explained more precisely, which concrete method of qualitative research was applied. Mention of the thematic, axial and selective coding (line 244) would point on the use of the grounded theory method, but the rest of the text doesn’t show the systematic application of this method. It is mentioned neither in the data collection or synthesis nor in the Bibliography.

The concrete description of application of the (participative?) observation method is missing. Author should provide information on the concrete way and length of this observation.

Each step of the algorithm of the data collection, analysis and synthesis should be described clearly to allow other researches to repeat it in their own research.

The research localities should be marked in the map (Figure 1).

The resources backing the mention of different meaning of prosperity for indigenous peoples (lines 215-219) are missing.

The description of the figures should be corrected: the Picture 1 = Figure 1 and its resource must be described in the standard way, not just giving the URL, Table 2 = Figure 2.

Author Response

Dear reviewer! 

Thank you very much for your comments. We have tried to take into account all the comments in our work.

1.       We expanded the theoretical framework. We used neo-institutional approach to explain social processes in Russian oil-dependent communities.  We gave a brief overview of the main ideas of this approach and analyzed the influence of oil resources on social relations in Russian society based on this concept.

2.       We described more carefully our research method. We used grounded theory for analysis of our data. We defined the main stages of the qualitative analysis and indicated the main analytical categories and subcategories which we operated.

3.       We provided more information about our participative observation. We pointed the information about the places, the period, and our activity during the observation.

4.       We indicated the different perceptions of well-being among local inhabitants (lines 219). It helped to explain the reasons of the conflicts in the oil-dependent communities.

5.       We corrected the description of the figures. We put the new version of the map of the Russian regions, which is clearer. It allows to see the location of the regions in which the study was conducted. But it is difficult to indicate the concrete villages on this map, because they are too small for the map scale.

6.       We did not make a professional language-check, because we had only for 5 days for corrections. But we are ready to do it further.

Round 3

Reviewer 2 Report

The quality of the manuscript is now appropriate, however, it still needs a slight revision.

In the chapter “Research materials and questions”, the phrase “the most frequent code” should be reconsidered, for the qualitative research the relevance of the meaning of the code for your research question is much more important than its frequency. The grounded theory method shouldn’t be used just for some of the research phases (data analysis) but in all the research process!  Otherwise, you should be explain, why it was used just for the selected part of the research.

The research localities should be really marked in the map (Figure 1), just indicatively (not concrete villages, just NAO, KMAO), YNAO and the Sakhalin Island), e.g. by red circles, it is possible. Also, the times of research conducted in different research localities are not described in the chapter “Research materials and questions”. It should be completed.

The description of the figure resource should be described in the standard way, e.g. by number of referenced resource described in the Bibliography, e.g. the footnote 2 (in the line 30) could be simply replaced by the referencing by number [3]. The same pays for the footnotes 12 and 13.

Also, I noticed that the article “Tysiachniouk, M., Tulaeva, S. Benefit-Sharing Arrangements between Oil Companies and Indigenous People in Russian Northern Regions. Sustainability 2017, 9(8), pp. 1326. Available online: 10.3390/su9081326 https://www.mdpi.com/2071-1050/9/8/1326/htm” is very similar to the submitted manuscript, so I recommend to reconsider (mainly in the conclusion chapter) stressing of the new contribution of this new manuscript (in the comparison with the previous one, published in Sustainability).

Author Response

Dear reviewers,

Thank you so much for your careful work with our article. We are sure it allows us to improve our paper. We tried to correct the article in accordance with your recommendations.

1.     We have corrected the methodological chapter. We used grounded theory on different stages of our work: collection empirical data, analyzing materials, comparison our new data with analytical categories, making theoretical conclusion, and validation of the results. We included it in the text.

2.     We pointed out the time of the research trips and collection of empirical data.

3.     We have marked studied areas in the map with the red color. 

4.     We have corrected the references and footnotes in accordance with the standard requirements.

5.     We have expanded the conclusion of the article. Compared to the previous article, the new paper analyzes broader social processes connected with the benefit sharing agreements between oil companies and local communities in Russian society.  It includes the development of informal exchanges between different groups of actors, ritualization of the public participation, oil patronage in social and cultural spheres. We described how the oil colonization transformed social norms and expectations in the communities. Special attention we paid the cultural programs supported by oil companies and its consequences for Indigenous culture.
